# Enzyme Models—From Catalysis to Prodrugs

**DOI:** 10.3390/molecules26113248

**Published:** 2021-05-28

**Authors:** Zeinab Breijyeh, Rafik Karaman

**Affiliations:** Pharmaceutical Sciences Department, Faculty of Pharmacy, Al-Quds University, Jerusalem P.O. Box 20002, Palestine; z88breijyeh@gmail.com

**Keywords:** enzymes, computational methods, catalytic models, intramolecularity, proton transfer reactions, prodrug approach

## Abstract

Enzymes are highly specific biological catalysts that accelerate the rate of chemical reactions within the cell. Our knowledge of how enzymes work remains incomplete. Computational methodologies such as molecular mechanics (MM) and quantum mechanical (QM) methods play an important role in elucidating the detailed mechanisms of enzymatic reactions where experimental research measurements are not possible. Theories invoked by a variety of scientists indicate that enzymes work as structural scaffolds that serve to bring together and orient the reactants so that the reaction can proceed with minimum energy. Enzyme models can be utilized for mimicking enzyme catalysis and the development of novel prodrugs. Prodrugs are used to enhance the pharmacokinetics of drugs; classical prodrug approaches focus on alternating the physicochemical properties, while chemical modern approaches are based on the knowledge gained from the chemistry of enzyme models and correlations between experimental and calculated rate values of intramolecular processes (enzyme models). A large number of prodrugs have been designed and developed to improve the effectiveness and pharmacokinetics of commonly used drugs, such as anti-Parkinson (dopamine), antiviral (acyclovir), antimalarial (atovaquone), anticancer (azanucleosides), antifibrinolytic (tranexamic acid), antihyperlipidemia (statins), vasoconstrictors (phenylephrine), antihypertension (atenolol), antibacterial agents (amoxicillin, cephalexin, and cefuroxime axetil), paracetamol, and guaifenesin. This article describes the works done on enzyme models and the computational methods used to understand enzyme catalysis and to help in the development of efficient prodrugs.

## 1. Introduction

The largest group of proteins are called enzymes, which are outstanding, highly specific biological catalysts that accelerate the rate of chemical reactions (>10^17^-folds) within the cell, and are classified according to the Enzyme Commission (EC) number into seven main groups: oxidoreductases, transferases, hydrolases, lyases, isomerases, ligases, and translocases [1,2,3]. Enzymes’ activity depends on several factors, such as pH, temperature, pressure, cofactors, and the availability of a substrate. The substrate binds to the enzyme by two different proposed models: the lock and key model, where the substrate fits perfectly into the active site of the enzyme and complement each other, and the induced fit model, where the substrate does not fit precisely and its binding induces the alignment and reshape of the active site [4,5]. The active site is a small functional area that lies in the core of the protein structure, which contains a hydrophobic binding pocket with three amino acid residues called the catalytic triad, histidine, aspartate, and serine, in most of the hydrolase enzyme. Additionally, there are nearby complementary residues such as peptide N–H moieties in the oxyanion hole (an arrangement of hydrogen bond donors). These moieties support the functional role of the active site residues in reducing the activation energy by participating in H-bonding with reaction intermediates and transition states [6,7]. The enzyme–substrate complex (ES) is formed due to the binding energy, induced fit, and several catalytic reactions at the active site, including (1) covalent catalysis, (2) general acid–base catalysis, (3) metal ion catalysis, and (4) catalysis by approximation, in which all work to lower the binding energy and stabilize the transition state [8,9]. The conversion of the substrate (S) to a product (P) in the presence of the enzyme can be illustrated as changes in energy; for the reaction to move forward, the substrate must pass the activation energy to be converted to higher energy (transition state). Enzyme increases the rate of the reaction by reducing the activation energy [10].

Understanding the enzyme’s mechanism of action to reach high-rate enhancement and specificity is essential in studying the biochemical processes that can help in the development of drugs and catalysts. The main challenge for the researcher is to mimic the same structural features of hydrolases in a synthetic catalyst system. Molecular simulations and modeling are very important in providing information about enzyme-catalyzed reactions where experimental research measurements are not possible. The most used methods for modeling the structure and dynamics of enzymes are molecular mechanics (MM) and quantum mechanical (QM) methods [11,12]. Numerous computer simulations of enzymatic reactions have indicated that the stabilization of the transition state is the main catalytic factor [13]. The path to achieving a stable transition state has split scientists into those like Warshel’s school, who believe that enzyme catalysis is a result of preorganized water molecules that stabilize the transition state and cause a reduction in the folding energy at the active site, and not due to the interaction between the enzyme and substrate [14,15,16], and others like Menger, Nome, and coworkers, who believe that spatiotemporal effects that are based on geometric factors are responsible for the catalytic effects of the enzymes, which are independent of any solvent reorganization considerations [17]. This review gives a summary of the computational methods and theories used to understand enzyme modeling and design promoieties to be attached covalently to active drugs for the development of novel prodrugs. Upon exposure to a physiologic environment, these prodrugs go through interconversion to a nontoxic moiety and the active parent drug in a programmable manner. The rate of drug release is dependent solely on the rate-limiting step of the interconversion reaction. In this approach, no enzyme is needed to catalyze the interconversion reaction.

## 2. Computational Methods

Understanding the mechanism of the catalytic power of enzymes is essential in the study of complex biological processes, drug design, and the development of novel catalysts in industrial processes. Experimental studies suffer from indirect and insufficient evidence for determining the detailed reaction mechanism, such as the structure of the transition state [18]. Computation, on the other hand, is an essential method for (1) information integration from multiple experimental approaches, (2) providing a detailed characterization of enzyme behavior and function, and (3) providing suggestions for new experiments and predictions of their outcomes. Despite that, computational methods cannot be completely reliable or predictive; therefore, for a deep understanding of enzyme mechanism and energetics, computational methods must be bolstered by recourse to experiment [19]. For the simulation of a complex reaction in a solution or enzyme, efficient computational tools, such as simple, empirical methods (e.g., molecular docking) or more complex methods that are based on the laws of physics (e.g., QM/MM) or free energy perturbation (FEP), are necessary to provide structure energy calculations for the prediction of potential drugs or prodrugs [20]. Table 1 summarizes the main differences between these methods and their applications. A small number of atoms participate in bond-forming or breaking processes in chemical reactions, while many other atoms do not undergo changes in electronic structure but influence the properties and reactivity of the active site by serving as a steric and electrostatic environment. The QM and MM approach was first developed by Warshel and Levitt, where QM simulates the active site of an enzyme at the electronic level, while MM simulates the rest of the system at the atomistic level [21,22]. QM and MM are reliable methods, but at the same time, they are money, effort, and time-consuming. QM calculations include ab initio, semi-empirical, and density functional theory (DFT) [23]. The ab initio method is based on the Schrodinger equation with some number of estimations and used in small molecules with a restricted number of atoms (not more than 30 atoms) for the calculation of electronic distribution and other features of molecules [24,25]. Semi-empirical methods are based on approximations to the Hartree–Fock (HF) theory, such as AM1, PM3, MINDO, MNDO, MINDO/3, and SAM1. These methods can be used to study larger systems and carry out longer sampling [26,27]. The density functional theory (DFT) method is a less demanding computational method that provides structural or energy calculations for intermediate-size biological systems. This method describes energetics and kinetics in terms of the properties of the reagents in the ground state to give insight into how enzymes work. DFT has been used successfully to study generalized acid–base reactions, inorganic complexation reactions, redox reactions, and pericyclic reactions [28]. On the other hand, MM is a mathematical approach used for the computation of various physical properties, such as structure, energy, dipole moment, and evaluating large biological systems, such as proteins, large crystal structures, and large solvated systems. However, this method is restricted by a vast number of torsion angle computations present in structurally diverse molecules [29,30]. QM/MM modeling of enzyme-catalyzed reactions has been used to develop and test theories of enzyme catalysis and study various intramolecular processes to gain knowledge about enzyme reaction mechanisms and to determine the factors affecting the rate-determining step. Using QM/MM modeling in the determination of enzyme reaction pathways and transition states generally helps in the design of drug leads and catalysts, in addition to understanding drug metabolism and resistance [31,32,33].

## 3. Enzyme Catalytic Models

Several chemical models have been presented by scientists to mimic the high rate of acceleration achieved by enzymes, including (1) Koshland in his “orbital steering” theory (Figure 1), which describes reactive atoms with at least one spherical asymmetry in their valence orbitals, which are constrained by binding to the active site or by the superstructure of the molecule in an intramolecular reaction to react along specific pathways. Koshland concluded that the orientation factor, which can be obtained from the contribution of rotational and vibrational entropy from transition states, plays an important role in the catalytic power of the enzyme. The orientation factor should be greater than 1 if the constraints lead the reacting molecules through an optimal pathway [40]. (2) Bruice, in his intramolecular cyclization of dicarboxylic semi-esters (Figure 1), deals with the driving forces for enzymatic and intramolecular reactions in a term called the near attack conformation (NAC) that define the required conformation for reactants to enter TS. The greater rate of constant depends on the mole fraction of NAC reactants, the change in reactant solvation within the NAC, and the electrostatic forces (e.g., hydrogen bonds, metal ligation). Bruice concluded that the driving force for the rate enhancement is under enthalpy control that arises from both ground-state and transition-state features and not entropy [41,42]. (3) Milstien and Cohen in their gem-tri-methyl lock (stereopopulation control) (Figure 1) studied the acid-catalyzed lactonization of some hydroxyhydrocinnamic acids. Their study showed that increasing the introduction of moderately large substituents such as methyl groups in the side chain and in aromatic ring size (six-membered lactone) leads to large enhancements in both equilibrium and rate constants for cyclization. This result was found to be due to the effect of bulky substituents, which produces a severe conformational restriction “freeze” of the side chain. The freeze effect accelerates ring closure to a greater extent, especially in the gem-dimethyl-methyl interaction, where the three skeletal bonds of the side chain are largely frozen into a conformation, which is highly suitable for the formation of the tetrahedral intermediate [43,44,45,46].

(4) Kirby’s enzyme models (Figure 2) provide an attractive way to understand the efficiency of enzyme catalysis by studying intramolecular reactions. Effective molarity (EM) (the ratio of intramolecular reaction rate constant divided by intermolecular process rate constant undergoing by the same mechanism and same conditions) has been measured for hundreds of intramolecular reactions; the higher the EM, the more efficient the intramolecular reaction, such as in the case of intramolecular cyclizations, to form five- or six-membered rings that involve addition and substitution reactions where EM is estimated to be between 10^3^ and 10^9^ and can reach 10^13^ M [49,50]. Kirby studied the efficiency of general acid catalysis in intramolecular model systems and proton transfer between two oxygen atoms, oxygen–nitrogen atoms, and from oxygen to carbon atoms in several systems, such as the hydrolysis of several substituted N-alkylmaleamic acids, and showed that the reaction rate is enhanced by 10^10^M or more when the carboxylate groups lie within the plane of the central double bond or when the attacking oxygen atom is closest to the amide carbon in a favorable angle approach. In addition, it was suggested that the rate-limiting step is the dissociation of the tetrahedral intermediate [51]. Further, Kirby’s studies on the hydrolysis of dialkyl acetals benzaldehyde, 1-arylethyl ethers of salicylic acid, enol ethers, and vinyl ethers and the nucleophilic attack on a phosphodiester revealed that the development of strong hydrogen bonds in the transition state is the main key feature of general acid–base catalysis in enzyme reactions [52,53,54,55,56].

(5) Menger, in his spatiotemporal hypothesis (Figure 3), suggested that intramolecular reactions are much faster than their intermolecular counterparts because they are oriented in a synthetic, physical, or biological way. Enzyme and its bounding substrate form an intramolecular organic process where they are holding their reactive groups in proximity via noncovalent and covalent bonds, respectively. Menger examined the intramolecular reactivity between two functional groups that are held by a rigid carbon framework at well-defined angles and distances. The study concluded that the rate of the reaction between two functional groups is proportional to the time that these groups reside within a critical distance with a requirement of energy to extrude the solvent between two reactants to move them into bonding distance. The distance separating nucleophile and electrophile is a critical parameter in nucleophilic reactivity, which is estimated roughly to be 2.8 Â, and when the distance falls below 2.7 Å, the activation energies become vanishingly small. For example, chymotrypsin hydrolyzes amides with 10^8^ accelerations, maybe because it holds the catalytic groups and the amide carbonyl at bonding distances with small general base catalysis [58,59,60]. Menger also examined hydrolyzing unactivated amides using MM calculations and found that the carboxyl oxygen lies within the van der Waals contact distance of the amide carbonyl carbon and is projected to cleave the amide at chymotrypsin-like rates. Experiments showed that a single carboxyl is necessary for amide cleavage, and the fast rate (EM = 10^12^ M, t_1/2_ = 8 min) arises from sustained proximity at van der Waals contact distances. Several studies conducted by Menger’s group using the protease model for the hydrolysis of amide bond showed the same results. Menger’s spatiotemporal hypothesis suggests that the enzyme system uses the “split-site” model where a substrate is divided into a binding portion (ES_B_) and a reactive portion (ES_R_). ES_B_ remains constant, while the ground state transforms into the transition state, and its interactions are assumed to stabilize the ES complex due to attractive forces, such as hydrogen bonding, hydrophobic bonding, electrostatics, and van der Waals association. In contrast, ES_R_ interactions are assumed to be destabilizing and are altered during catalysis due to substrate chemical modifications and the desolvation of the catalytic groups to reach proper contact with the substrate. The attractive interactions at the binding site accelerate the enzyme reaction, which can be due to (1) the ground-state effect because the rate enhancement is achieved by binding more tightly to the ground state via a hydrogen bond, or (2) the transition-state effect because the new hydrogen bond lowers the transition energy. Both explanations are acceptable because any changes in the ground state are translated into an identical effect in the transition state [61,62,63,64,65].

## 4. The Effective Molarity (EM)

Enzyme efficiency in the catalysis of biochemical reactions is dependent on the selective binding and the stabilization of the transition state. Effective molarity (EM) reflects rate enhancements, which are defined as the ratio between the intramolecular rate constant (*kintra*, unimolecular) and the bimolecular process (*kinter*). EM values depend on different factors, including ring size, solvent, and reaction type, and can range from 0.3 M to more than 10^16^ M, as reported by Kirby. EM measurements require the same mechanism of intra- and intermolecular processes with the same reaction conditions for rate measurements. Due to these restricted conditions, only a few reactions have their known experimental EM values; as a result, alternative ways were explored to obtain EM values for other important intramolecular processes. Ab initio and DFT calculation methods were used by Karaman to calculate the EM values for several intramolecular SN2 processes, such as substituted 3-aminoalkyl halides to substituted aziridines and substituted chlorohydrins to substituted epoxides. The results showed that the proximity orientation is the driving force for ring-closing reactions in the two systems alongside the ground strain energies of the products and the reactants. Besides, a strong correlation between calculated and experimental EM values was found, which opens a door for predicting the EM values for processes that are difficult to be obtained experimentally [67]. Moreover, using ab initio and DFT calculations, an equation that correlates activation energies with effective molarity was established from the studies on ring-closing reactions of three-, four, five-, and six-membered rings. The results showed that the substitution effect in five-membered rings is much dominant than that in the three- and four-membered rings, and increasing the size of the ring formed and the volume of the nucleophile involved in the ring closing decreases the need for directional flexibility [68]. Similar results were obtained from the study on Brown’s system (4-bromobutyl alcohols and 4-bromobutyl amines), which demonstrated that the strain effect and not the proximity orientation is responsible for the acceleration in the rate of ring closing, which is also strongly correlated with the distance of two reactive centers and the attack angle [69,70]. Proton transfer reaction’s rate was also confirmed to be dependent and linearly correlated with the distance between the two centers as well as with the EM values [71]. Additonally, computational studies on the ring closing of enol ethers and amine olefin (aldolase, isomerase, and ammonia-lyase Kirby’s enzyme models) demonstrated that a proton transfer is a rate-determining step that involves two stages: (1) preorganization of the global minimum structure from low enthalpic energy and long distance between two reactive centers to a more organized structure with a shorter distance, which depends on the proximity effects, and (2) transfer of the proton from the organized stage to the transition state, which is largely affected by the strain energies of the reactant and product [72,73]. Similar results were found with proton transfer reactions in Kirby’s acetals, which occur via intramolecular general acid catalysis (IGAC), and the driving forces of this process are the distance between two reacting centers, the attack angle, and the hydrogen bond strength between the carboxyl and acetal ether oxygen groups. This is in perfect agreement with Menger’s spatiotemporal hypothesis [57]. Moreover, using computational DFT calculation methods at B3LYP/6-31G (d,p), B3LYP/311+G (d,p) levels, the mechanism of intramolecular acid-catalyzed hydrolysis of Kirby’s N-alkylmaleamic acids was studied, and the results indicate that the reaction goes through three steps: proton transfer, nucleophilic attack, and intermediate dissociation to provide products. Additionally, it was found that the rate-limiting step is dependent on the reaction solvent (water or gas phase), and the rate of hydrolysis is linearly correlated with the strain energy of the tetrahedral intermediate or the product. Furthermore, DFT methods demonstrated great credibility in predicting energies and rates for reactions with a linear correlation between calculated DFT EM values and experimental EM values for the examples given herein [74]. To analyze the contribution of hydrogen bonds to the efficiency of proton transfer in intramolecular processes (rigid intra-and intermolecular model systems), DFT at B3LYP/6-31G(d,p) calculations were conducted, and the results revealed that high EM values (>10^8^) are achieved when both the transition state (TS) and the product geometries are favorable, such as in pathways involving the reactions of nucleophiles in which the distance and orientation of angle of attack are short and close to linearity, respectively. Then the effect of strain release through reaction process and geometry changes involved parts of the reacting system on the free energy at the transition state. On the other hand, low EM values were noticed in intramolecular proton transfers that are associated with general acid or base catalysis (IGAC-IGBC), which is explained by the loose character of the TS that does not need a full loss of motion of reacting groups to allow proton transfer in addition to specific features that are correlated with the low mass of proton [75,76]. Using DFT calculations at the ωB97X-D functional with the 6-311++G (d,p), Scorsin et al. examined 2-carboxyphthalanilic acid (2CPA) hydrolysis reaction, which includes both proton transfer and nucleophilic attack, and found that 2CPA is very prone to the N-protonation reaction path, which involves general acid proton delivery to the amide nitrogen by the carboxyl, engaging of another carboxylate in the nucleophilic attack, unchanged amide carbonyl in the transition state, two bond-forming events, and spatiotemporal-base rate acceleration. On the other hand, both the general base catalysis of carboxylate coupled to the general acid catalysis of carboxyl and the nucleophilic attack of carboxylate in the amide carbonyl that is coupled to the general acid catalysis of amide oxygen are not operative [77].

## 5. Driving Forces for Rate Accelerations in Some Intramolecular Processes

Several studies were conducted to explore the driving forces affecting the remarkable acceleration rates in several intramolecular reactions using different computational methods, such as ab initio molecular orbital at different levels, DFT, semi-empirical molecular orbital, and MM methods. These methods were used to study (1) the thermodynamic and kinetic behavior of the lactonization of some hydroxy acids (studied by Cohen and Menger), (2) the SN-2-based cyclization reactions of Bruice’s di-carboxylic semi-esters and SN-2-based ring-closing reactions by Brown and Mandolini, (3) the cyclization of some *ω*-bromoalkanecarboxylate anions, and (4) the proton transfer reactions in Menger’s system and Kirby’s enzyme models. Studies revealed that the enhancement in the proton transfer process is largely dependent on the bond distance and angle of attack between two reacting centers. This was achieved by expanding the equation that relates the rate of acceleration with distance derived by Menger to a new equation that relates rate with both the angle and distance; the equation combines Menger’s and Koshland’s hypotheses, where neither distance alone nor angle of attack alone is the dominant force that enhances the rate in intramolecular reactions [66]. Menger in his spatiotemporal theory and Bruice in his “near attack conformation” concept highlighted the importance of distance in intramolecular reactivity. Despite the huge attention to intramolecularity, the detailed relation between intramolecular reaction rates and distance/angle remains unclear; therefore, connecting the activation energy with the distance between two reactive centers and the hydrogen bond angle can provide an excellent tool for reaction rate prediction. This concludes that enzymes may achieve their exceptional catalytic activity at the active site by forcing a specific range of contact distances within the space of hydrophobic pockets [78]. The DFT calculations conducted on the lactonization of the tri-methyl lock system (acid-catalyzed and uncatalyzed lactonization reactions) revealed that the rate enhancement in lactonization is largely the result of a proximity orientation and not strain effect [79]. Besides, the intramolecular cleavage reaction of mono- and di-acid amides (Kemp’s amidase enzyme models) was researched by DFT calculations at the B3LYP/6-31G (d, p) level, and the outcome of the study revealed that the rate of enhancement for the cleavage reactions in amides under neutral pH range is due to proximity orientation of the carboxylic proton and the amide carbonyl oxygen, which is the rate-limiting step in this process. On the other hand, the contribution of the ground-state pseudoallylic strain effect is little or negligible [80]. Similar results were obtained from the HF/6-31G and B3LYP/6-31G (d,p) calculations of the amide cleavage reactions of Menger’s di-carboxylic aliphatic amides (peptidase enzyme model) and Bender’s aromatic amide [81]. The acceleration observed in the cyclization of di-carboxylic semi-esters in Bruice’s system has been computationally studied, and the HF/6-31G, HF/6-31G (d,p), and DFT at B3LYP/6-31G (d,p) calculations demonstrated that the activation energy is dependent on the strain energies of the transition states and the reactants and not on the proximity orientation, in contrast to that suggested by Bruice [48].

The above computational studies have revealed the following conclusions: (1) the rate enhancement in intramolecular reactions is a result of proximity orientation where its source can strain effects or not strain effects; (2) the significant rate enhancements in intramolecular reactions are caused by both enthalpic and entropic effects and not only due to enthalpic effects, as proposed by Bruice; (3) the nature of the reaction, intermolecular or intramolecular, is greatly determined on the distance between the two reactive centers; and (4) proton transfer between two oxygens or oxygen and nitrogen in Kirby’s acetal systems is the result of strong hydrogen bonding developed in the product and the transition states leading to them.

## 6. The Prodrug Approach

Prodrugs are pharmacologically inactive substances that are metabolized in the body into their corresponding active drugs. The prodrug approach is used to overcome negative pharmacokinetic properties of a drug (problems with solubility, absorption, and distribution; instability; toxicity, site specificity; formulation problems such as unacceptable bitter taste sensation; and others) and to optimize clinical profile. Prodrugs may contain one or two promoieties that are attached to a parent active drug molecule and can be cleaved by enzymatic, chemical reaction, or by molecular modification such as oxidation/reduction reactions [82,83,84,85].

### 6.1. Enzyme-Mediated Prodrug Activation

Several enzymes are involved in the activation of prodrugs, such as oxidoreductases such as CYP450, and hydrolytic enzymes (e.g., carboxylesterase, butyrylcholinesterase, acetylcholinesterase, alkaline phosphatase (ALP), phospholipase A_2_ (PLA_2_), and human valacyclvirase) [23]. CYP450 enzymes such as CYP3A4 and 2C9 are the main enzymes responsible for drug metabolism and activation of prodrugs in which CYP3A4 is accountable for the oxidation of approximately two-thirds of all known drugs and CYP2C9 is responsible for the metabolism of approximately 15% of drugs on the market [86]. Examples of prodrugs activated by CYP450 are anticancer drugs such as cyclophosphamide and ifosfamide, which are activated by 4-hydroxylation mostly by CYP2B6 and CYP3A4, respectively [87,88,89,90,91,92], in addition to other prodrugs that target other clinical conditions such as cardiovascular disorders, inflammation, and allergies [93].

Polymorphism in CYP450 enzymes can affect enzymatic activity; for example, polymorphism of CYP2C9 has shown to decrease enzymatic activity such as CYP2C9.30, which is found in Japanese people, which is responsible for decreasing the activity of prodrug losartan. Computational methods such as MD simulations were used to elucidate the mechanism of amino acid replacement, which affects drug metabolism and the regioselectivity of CYP450. MD simulations found that mutation in the CYP2C9.30 variant causes rigidity and changes in the substrate fitting to the active site, decreasing its access to some protein channels, which could be responsible for altered catalytic enzymatic activity [94,95,96].

On the other hand, hydrolytic enzymes such as phospholipase A_2_ (PLA_2_) are responsible for the hydrolysis of phospholipid-based prodrug at *sn*-2 positioned fatty acid; therefore, conjugation of phosphate or glyceride to the drug results in the enzymatic activation of the designed prodrug and the liberation of the free drug moiety. Activity of PLA_2_ towards phospholipid-based prodrugs can be determined by different molecular modeling methods such as molecular docking and MD simulations to point out the structural adjustments of the conjugated moiety and their length that are required to get the highest degree of prodrug activation [97,98]. Another example of hydrolytic enzymes are human carboxylesterase (hCE) types 1 and 2 that hydrolyze prodrugs with ester bonds such as epalrestat and natural antioxidant products used for diabetes complications. The hCE catalytic site contains serine, histidine, and glutamine, where the hydroxyl group in the serine attacks the carbonyl group of the ester prodrug, and the rest of the amino acids stabilize the complex [23,99]. Several computational calculations were performed to highlight the prospect of hydrolytic cleavage of the ester prodrugs via esterase, such as QM factors (e.g., highest occupied molecular orbital (HOMO)—lowest unoccupied molecular orbital (LUMO) energy gap), which provide an insight into electron transfer from protein to the prodrug within the catalytic domain by using Schrödinger software, and DFT analyses. Geometrical parameters such as Burgi–Dunitz angle and distance were also used [23,99,100].

However, often the activation enzymes are unidentified, and only a limited number have a three-dimensional structure determined, in addition to the high costs of enzyme production, which makes them unfavorable from an economic point of view.

### 6.2. Chemical Approach for Prodrug Activation

The chemical approach in which the parent drug is linked to a linker and the targeted drug approach such as virus-directed enzyme prodrug therapy (VDEPT), antibody-directed enzyme prodrug therapy (ADEPT), and gene-directed enzyme prodrug therapy (GDEPT) are the main two prodrug approaches [101,102,103,104]. The two main classes of the prodrugs’ chemical approach are (1) the carrier-linked prodrugs, which can be classified into bipartite prodrugs, where the carrier is linked to the parent drug directly; tripartite prodrugs, where the carrier is attached to the parent drug through spacer links; and mutual prodrugs, which consist of two drugs that are linked together chemically and act as promoiety to the other, and (2) bioprecursors, which are inactive substances without a carrier and converted rapidly to the active parent drug after metabolic reactions (Figure 4) [105,106,107,108].

Several prodrugs based on the chemical approach have been developed in the past few decades, such as the following: (a) ester prodrugs owe in vitro chemical stability and susceptibility to esterases that result in a rapid conversion to the parent drug once it enters the body. Ester prodrugs are used to improve lipophilicity and increase membrane permeation by masking the charge of polar functional groups such as ibuprofen guaiacol ester, acyclovir aliphatic ester prodrugs, thioester of erythromycin, and palmitate ester of clindamycin [101,109,110,111,112,113,114]. (b) Amide prodrugs enhance stability, change lipophilicity, and provide targeted drug delivery. Drugs with a carboxylic acid or amine group can be converted into amide prodrugs by an amide linkage that undergoes fast amide hydrolysis in vivo by the action of nonspecific amidases or by specific enzymatic activation (e.g., renal γ-glutamyl transpeptidase) [115,116]. (c) Carbonate and carbamate prodrugs are more stable than their corresponding ester prodrugs and less stable than the corresponding amides, and have no specific enzymes for their hydrolysis and are generally degraded by esterases [117,118]. (d) Oxime prodrugs increase the permeability of the active drug and are converted by the action of microsomal cytochrome P450 enzymes (CYP450). An example of this approach is the dopaminergic prodrug 6-(*N*,*N*-Di-*n*-propylamino)-3,4,5,6,7,8-hexahydro-2*H*-naphthalen-1-one [119,120]. (e) Imine prodrugs (N-Mannich, enaminones, and Schiff bases) are used to enhance drug solubility, such as rolitetracycline, a Mannich derivative of tetracycline; cleavage of these prodrugs is strictly pH dependent [115]. (f) Phosphate and phosphonate prodrugs increase the aqueous solubility, such as in the case of prednisolone sodium phosphate and fosamprenavir, and are cleaved to give the parent drug by alkaline phosphatases in the gut [121,122,123,124]. (g) Azo compounds are used for targeted drug release by the action of azo reductase from colonic bacteria. Examples are sulfasalazine and osalazine [125,126]. (h) Polyethylene glycol (PEG) conjugates are used to enhance the solubility and prolong drug plasma half-life. Several linkers can be used to attach the drug to PEG, such as ester, amide, carbamate, or carbonate spacer, which can be cleaved by enzymatic action. An example is a doxorubicin conjugated to PEG [127,128].

Classical prodrug approaches focus on alternating the physicochemical parameters, while modern approaches are based on intramolecular processes utilizing computational methods and correlations between experimental and calculated values to design linkers to be attached covalently to the active drug. The designed prodrug will undergo programmed interconversion to a nontoxic moiety and the active parent drug upon exposure to the physiologic environment. Such linkers are those described in the cyclization reactions of di-carboxylic semi-esters in Bruice’s enzyme model, proton transfers between two oxygens in Kirby’s acetal model, acid-catalyzed hydrolysis of Kirby’s N-alkylmaleamic acids, and Menger’s rigid carboxylic amides. In this approach, the rate of prodrug release is dependent on the rate-limiting step of the interconversion reaction, and no enzyme is needed for the conversion [129,130,131,132,133,134,135,136]. Additionally, Cohen’s model was used to design a chemically driven pro–prodrug where a pharmacologically inactive moiety such as hydroxyhydrocinnamic acid was attached to low water-soluble drugs to enable their intravenous administration. AM1 semi-empirical molecular orbital and ab initio at HF/6-31G level calculations for uncatalyzed and acid-catalyzed lactonization of different hydroxy acids revealed the following: (1) The rate-limiting step for lactonization of hydroxyl acids is not the collapse of tetrahedral intermediates as suggested by Houk et al. but the formation of tetrahedral intermediates, which is composed of two steps: first, the approach of a hydroxyl group to the carbonyl carbon and then the proton transfer from etheric oxygen to the anionic oxygen of the carboxylic moiety. (2) The value of the activation energy can be predicted by using AM1 and the HF calculations if the distance between the reacting centers and the angle of the attack is known. (3) Thermodynamic properties are necessary for the identification of the transition state and the calculation of the free energy of activation and not for the estimation of kinetic rates as suggested by Wilcox et al. [47]. Prodrugs were also used to examine the concept of strain energy to understand the rate of acceleration of several enzyme models. Strain energy, specifically the ring strain energy, has been assumed to be the main driving force for rate and reactivity enhancements. The enzyme models of different ω-bromoalkanecarboxylate anions were studied by using B3LYP/6-31G (d, p) and B3LYP/cc-pVDZ levels, and introduced a new equation that correlates ring-closing rate with the strain energy and the distance between two reactive centers. These results may help in predicting strain energies of small, medium, and large ring-closing reactions and assist in designing prodrug systems that can help in improving the biopharmaceutical profile of different medications to enhance their effectiveness and deliver the active drug in a controlled fashion [137].

### 6.3. Masking Bitterness of Drugs

The sense of taste in humans has five basic taste qualities: bitter, sweet, salty, sour, and umami or savory tastes [138]. Bitter taste receptors are found on the posterior part of the tongue; humans have 25 bitter taste receptors encoded by the TAS2R genes. Several drugs, over-the-counter (OTC) preparations, and phytochemicals contain bitter taste active ingredients. Examples of bitter taste drug include pseudoephedrine, phenylephrine, dextromethorphan, prednisolone, dyphylline or diprophylline, chlorhexidine, atorvastatin, loperamide, terfenadine, salbutamol, guaifenesin, and amoxicillin, which is the most common antibiotic prescribed among children [136,139,140,141,142]. Children and infants prefer the sweet taste and reject bitter taste substances, which may affect their compliance and acceptance of medications and increase the risk of avoidable side effects, such as suboptimal dosing, which may result in persistent symptoms and frequent doctor visitation or even hospitalizations [143,144,145]. Therefore, it is important to find ways to mask the bitterness within the pharmaceutical formulation. Several approaches have been used to mask the undesirable taste and thus solve patient compliance. These include conventional taste-masking methods, such as using sweeteners, amino acids, and flavoring agents, which is ineffective alone in the case of highly bitter drugs, especially those given in high doses, compared with more efficient developed approaches, such as using lipophilic vehicles (lipids and lecithin), liposomes, coating adsorbents, microencapsulation, ion exchange resins, and pH modifiers [146,147]. These approaches have helped in masking some drug formulations, but a serious challenge still faces pharmacists in masking the bitter taste of pediatric and geriatric formulations. Therefore, two new novel approaches have been designed (1) bitterless prodrugs that attach a promoiety to the active drug to bind to the bitter taste receptor and (2) bitter taste antagonists, which can be achieved by modifications to the structure and size of the bitter compound. Computational methods were used to investigate several intramolecular processes for the design of an efficient chemical device to be linked to a bitter drug such that the resulting prodrug cannot bind to the active site of the bitter taste receptor (thus masking the bitter taste), and upon passing from the oral cavity to the stomach, it interconverts to the bitter parent drug [139,148,149].

Guaifenesin (guaiacol glyceryl ether) (Figure 5, **1**) is an expectorant OTC medicine used in cough and cold preparations [150], but it has a strong bitter taste that makes it unacceptable for pediatric and geriatric patients. To mask its bitter taste, Karaman’s group used the prodrug approach in which guaifenesin binds to a promoiety (mono- and di-esters) (Figure 5) to block the functional groups responsible for the interactions with the cognate bitter taste receptor (hydroxyl group). The guaifenesin prodrug undergoes cleavage when exposed to the stomach and releases the parent drug and the nontoxic linker. Several ester prodrugs of guaifenesin (dimethyl maleate, maleate, glutarate, succinate, and dimethyl succinate) were designed by utilizing molecular orbital methods, which showed that the hydrolysis efficiency of the prodrugs is significantly sensitive to the distance between the nucleophile and the electrophile and to the pattern of substitution on the C=C bond. Calculations at the B3LYP/6-31G (d,p) level were performed for the ring-closing reactions of di-carboxylic semi-esters to explain the transition- and ground-state structures in water and in the gas phase. In addition, this highlights the importance of the orientation of the carboxylate anion to the ester carboxyl moiety in affecting the mode and rate of the ring-closing reaction. DFT calculations showed that the global minimum structures (GM) for all prodrugs exist in the condensed conformation where the distance between the nucleophile (O1) and the electrophile (C6) is shorter. Kinetic studies on the hydrolysis of the synthesized guaifenesin prodrugs at different pHs (3, 5, and 7.4) demonstrated sufficient stability at neutral pH values and rapid release under low pH conditions. Bitterness experiments revealed that the succinate derivative resulted in complete loss of TAS2R14 receptor responses [151].

Paracetamol (Figure 5, **2**) is an analgesic and antipyretic drug. Suppositories and syrups are the most used dosage forms in infants and children. Paracetamol has a bitter taste, which forms an obstacle for its administration to pediatrics and geriatrics where a large amount of the drug (from some hundreds of milligrams to 1 g) is needed for one dose; hence, it is important to mask the bitter taste of paracetamol upon administration to achieve good compliance [152,153,154]. In a comparison of paracetamol with phenacetin and acetanilide, which have a very slightly bitter taste and bitterless taste, respectively, it was found that the para position of the benzene ring is the only difference among these three compounds, where paracetamol has a hydroxyl group, phenacetin has an ethoxy group, and acetanilide has a hydrogen atom. This suggests that the hydroxyl group on the para position of the benzene ring is responsible for the bitter taste of paracetamol. Utilizing the proton transfer approach of Kirby’s enzyme model, a linker that blocks the phenolic group of paracetamol has the potential to eliminate the bitter taste of the analgesic drug. Three paracetamol prodrugs (Figure 5) with different linkers were designed such that they contain both hydrophilic (carboxylic acid group) and lipophilic moieties to ensure moderate hydrophilic–lipophilic balance (HLB). DFT calculation of the prodrugs revealed that the rate of proton transfer is dependent on the distance between the two reactive centers and the angle of attack. In addition, the linear correlation of the experimental and calculated EM values gives a good basis for designing paracetamol prodrugs that can mask the bitter taste of the drug and are capable of releasing the drug in a slow-release fashion, and this release is dependent on the type of the linker used [155].

Several marketed antibacterial drugs intended to be used orally, such as amoxicillin, cephalexin, and cefuroxime axetil, suffer from low stability when formulated as a suspension, and possess bitter taste, which reduces their compliance by children and geriatrics. The bitter taste is believed to be the result of a hydrogen bonding between the free amino group in the antibiotic drugs and the amino acids in the active sites of the bitter taste receptors. For example, cefuroxime axetil’s (Figure 6, **3**) bitter taste arises from the formation of hydrogen or ionic bond of the amido group at position 3 and the active site of the bitter taste receptors. Additionally, the drug suffers from relatively low bioavailability of 25% to 52%. Based on the acid-catalyzed hydrolysis of several maleamic acid amides, four different cefuroxime prodrugs (Figure 6) were designed using DFT calculations. The calculations revealed that the calculated t1/2 for the conversion of these prodrugs ranged between 12 and 200 min and that the reaction rate-limiting step was determined on the nature of the amine leaving group [156]. Amoxicillin (Figure 6, **4**) and cephalexin (Figure 6, **5**) suffer from low stability in aqueous media, where they might undergo hydrolysis when they are standing in solutions due to the reactivity of the strained lactam ring in which the carbonyl group undergoes nucleophilic attack by water to form the inactive penicilloic acid. Besides, both drugs have a bitter taste, which results in poor patient compliance, especially in pediatric and geriatric formulations. Based on Kirby‘s enzyme model, two linkers have been used for the design of novel amoxicillin and cephalexin prodrugs (Figure 6). This is to mask the bitterness of the parent drugs and to afford chemical devices with the potential to release the antibacterial agent in a controlled manner. QM calculations for those prodrugs revealed that the acid-catalyzed hydrolysis efficiency is significantly sensitive to the pattern of substitution on the carbon–carbon double bond and nature of the amine leaving group. Additionally, the calculations showed that antibacterial prodrugs can exist as a free carboxylic acid form in the stomach and as a carboxylate anion form in the blood circulation system and can undergo hydrolysis in an acidic aqueous medium while being stable at pH 7.4. The calculated half-life of the prodrugs was found to be largely affected by the pH medium. Moreover, kinetic studies have shown that the synthesized amoxicillin and cephalexin prodrugs do not bind to bitter taste receptors due to the presence of maleic or succinic promoiety on the amine group of the parent drug, which hinders the ability of the prodrugs to interact with the active sites of the bitter taste receptors [157,158].

### 6.4. Bioavailability Enhancement

Atenolol (4-(2-hydroxy-isopropylaminopropoxy)-phenylacetamide) (Figure 7, **6**) is a cardioselective beta1-adrenergic receptor antagonist used for the treatment of hypertension, angina pectoris, and cardiac arrhythmias. Atenolol is available as a tablet dosage form for oral administration and not formulated for easy administration to children because of its instability in aqueous solutions, so the preparation of liquid formulation of atenolol remains a challenge [159,160]. The prodrug approach (Figure 7) is used to enhance atenolol’s pharmacokinetic properties and bioavailability since the atenolol drug is a very hydrophilic compound that undergoes ionization in the stomach and intestine, and might undergo first-pass metabolism. Lipophilic atenolol prodrugs can provide a sustained-release device in a more convenient dosing regimen and better stability in aqueous solutions with fewer side effects. Linkers based on the acid-catalyzed cyclization reactions of Kirby’s model (N-alkylmaleamic acids) may serve as good carriers to atenolol. Atenolol prodrugs have an amide moiety, which makes them more stable compared with the parent drug with a free amine group. DFT calculations showed that the rate-limiting step in these systems depends on the reaction media. In the aqueous medium, the rate-limiting step of the cyclization is the collapse of the tetrahedral intermediate, whereas in the gas phase, it is the formation of the tetrahedral intermediate. Additionally, the acid-catalyzed hydrolysis is very sensitive to the pattern of substitution on the carbon–carbon double bond, and the rate of atenolol release is determined on the nature of the prodrug’s linker [161].

Dopamine (Figure 7, **7**) is a monoamine catecholamine neurotransmitter that activates the dopamine receptors, G protein-coupled receptors (GPCRs). Dysfunction of dopamine neurotransmission and its receptors leads to several pathological conditions such as Parkinson’s disease and hyperprolactinemia [162]. Dopamine is a water-soluble hydrophilic drug and cannot cross the blood–brain barrier. That is why its oral administration as a treatment for Parkinson’s disease is limited. L-Dopa (levodopa) is a dopamine direct precursor and can cross BBB to deliver dopamine. Levodopa is coadministered with carbidopa, a decarboxylase inhibitor that decreases the peripheral breakdown of L-Dopa and avoids the drug’s systemic side effects [163]. Therefore, the main approach was to develop new dopamine prodrugs (Figure 7) with higher bioavailability than the current medications to treat Parkinson’s disease by utilizing Menger’s enzyme model on the cleavage of some Kemp’s acid amides to the corresponding amines and anhydrides. A carboxylic group and hydrocarbon skeleton were used as hydrophilic and lipophilic moieties, respectively. The result showed that the rate of acceleration in Kemp’s model is dependent on the proximity of the carboxylic moiety and the amide carbonyl oxygen, and the difference in the activation energy values between the secondary amides and the tertiary amide is due to both proximity orientation and strain effects and not to pseudoallylic strain. Besides, DFT calculations predicted the half-life time (t½) of prodrugs’ conversion to release the parent dopamine drug to be between 12 and 20 h at pH 6 and higher at pH 7. The proposed prodrugs are believed to be more effective than L-dopa and can be used in different dosage forms due to their potential solubility in organic and aqueous media [164].

Statins are hydroxymethylglutaryl-CoA (HMG-CoA) reductase inhibitors that are used for the treatment of hypercholesterolemia, hyperlipoproteinemia, and hypertriglyceridemia. The HMG-CoA reductase enzyme is responsible for the conversion of HMG-CoA to mevalonate, a precursor of cholesterol [165,166]. Simvastatin (Figure 7, **8**) has a retarded aqueous solubility, low bioavailability (5%) mostly due to the extensive first-pass effect, and poor absorption rate from the gastrointestinal tract (GIT); therefore, improvement of its solubility and dissolution rate can enhance its bioavailability. The development of more hydrophilic prodrugs to release simvastatin in physiological environments such as the intestine is a promising strategy. DFT calculations of Kirby’s enzyme models were used to design simvastatin prodrugs (Figure 7) by linking the statin-free hydroxyl group to an amine linker (Kirby’s enzyme model). The calculations demonstrated that the reaction rate is linearly correlated to the distance between the two reacting centers and the angle of hydrogen bonding, and the rate-limiting step is the transfer of a proton from the ammonium moiety to the acetal oxygen. Additionally, the interconversion of the simvastatin prodrug can be determined according to the promoiety (Kirby’s enzyme model) structural features [167].

Phenylephrine (Figure 8, **9**) is a vasoconstrictor that possesses both direct (α1-adrenergic receptor agonist) and indirect sympathomimetic effects and is used for the temporary relief of nasal congestion by limiting the amount of fluid entering the nose and throat and decreases inflammation of nasal membranes. Besides, it is used for increasing blood pressure in adults with vasodilation in cases of septic shock or anesthesia and OTC eye drop formulations for mydriasis and vasoconstriction of the conjunctival blood vessel. Phenylephrine is absorbed and distributed rapidly into peripheral tissues but with a short elimination half-life (~2.5 h) and a relatively low oral bioavailability (~38%) compared with IV administration due to a high first-pass metabolism in the intestinal wall [168,169,170,171,172]. This may be due to the high polarity of phenylephrine; therefore, the development of a more lipophilic prodrug by blocking the phenolic hydroxyl group using Kirby’s linker may decrease the drug polarity and inhibit its first-pass effect metabolism. Several prodrugs of phenylephrine (Figure 8) were designed to enhance the bioavailability and tissue permeability and mask the bitterness of the parent drug and be cleaved chemically and not enzymatically in the intestine in a sustained-release manner. DFT calculations of the prodrugs showed that the rate-limiting step is the transfer of a proton from the carboxylic hydroxyl group to the neighboring acetal oxygen. In addition, the driving force of proton transfer is largely dependent on the geometric variations (the distance between the two reactive centers and the angle of attack), and the half-life for the conversion of the prodrugs to the parent drug can be programmed according to the nature of the prodrug linker [173].

Bruice’s and Kirby’s enzyme models were used to enhance the hydrophilicity and bioavailability of the antimalarial agent atovaquone (Figure 8, **10**), which was estimated to be less than 10% under fasted condition due to the lipophilic character of the drug. The prodrugs (Figure 8) were designed via the attachment of water-soluble carriers such as dicarboxylic semi-ester and the enol of the parent drug. DFT B3LYP ⁄ 6-31G (d,p) calculations showed that the rate of proton transfer in the designed prodrugs is responsive to geometric disposition (distance between two reactive centers and the angle of attack); short distance and minimal angle of attack give stronger intramolecular hydrogen bonding. Furthermore, result showed that the rate of enhancement in the cyclization of di-carboxylic semi-esters is the result of strain effects and not proximity orientation. The overall results revealed that the prodrug release rate is solely determined by the nature of the linker [174,175,176,177].

Tranexamic acid (Figure 8, **11**) is a synthetic lysine derivative and antifibrinolytic agent that prevents bleeding complications by blocking lysine binding sites on plasminogen molecules and inhibiting plasminogen interaction by forming plasmin and fibrin. Tranexamic acid is available in IV and oral dosage forms with reported bioavailability of 33%–34%. This may be due to the fact that tranexamic acid is an amino acid derivative and undergoes ionization in physiologic environments [178]. Therefore, designing more lipophilic tranexamic acid prodrugs (Figure 8) can provide the parent drug in a sustained-release manner and enhance its oral pharmacokinetic properties. Based on the proton transfer reaction in the acid-catalyzed hydrolysis of N-alkylmaleamic acids of Kirby’s enzyme model, DFT calculations at B3LYP/6-31G (d,p) and B3LYP/311+G (d,p) levels were utilized to design tranexamic acid prodrugs and to elucidate the transition-state and ground-state structures (reactants, intermediates, and products) in water and gas phases. The calculation results showed that the reaction rate-limiting step depends on the reaction medium, the nature of the amine leaving group, and the pattern of substitution on carbon–carbon double bond. Additionally, kinetic studies have shown that the half-life of the prodrug interconversion is dependent on the pH of the medium. At a physiological pH of 5.5–6.5, the carboxylic group of the prodrugs was converted to the corresponding carboxylate anion and yielded the parent drug, tranexamic acid, and the inactive linker as a by-product [179].

Azanucleosides (AZNs) are pyrimidine analogs of cytidine nucleoside and potent inhibitors of DNA methylation. Azacitidine, decitabine, and cytarabine are AZNs used in clinics as anticancer agents for the treatment of acute leukemias and myelodysplastic syndromes at lower doses [180,181]. Nucleosides suffer from high first-pass metabolism when taken orally and have relatively short terminal elimination (t1/2) [182,183,184]; therefore, designing slow-degrading prodrug or less hydrophilic aza-nucleoside prodrug systems in different dosage forms to release the parent drug in a controlled manner can result in better absorption of the drug via a variety of administration routes, enhanced clinical outcomes, more convenient dosing regimens, and fewer side effects. The decitabine (Figure 9, **12**) prodrugs were designed based on DFT B3LYP/6-31G (d,p) level calculations of Kirby’s enzyme models with hydrophilic moiety (N,N-dimethylaniline group) and lipophilic moiety (the rest of the prodrug). This might lead to better bioavailability due to the improvement of permeation. Kirby’s model carriers were selected due to the fact that they undergo proton transfer reaction to yield an alcohol, an aldehyde, and a hydroxyl amine. The rate-limiting step was found to be the transfer of a proton from the anilinium group into the neighboring ether oxygen, which is strongly dependent on hydrogen bonding in the product and transition state that lead to them. Additionally, the reaction rate depends on geometric disposition (distance between the two reactive centers and the angle of attack) and the structural features of the prodrug linker (Kirby’s enzyme model). Additionally, due to the unique structural features of the designed prodrugs that possess hydrophilic and lipophilic moieties and are capable of being soluble in organic and aqueous media, the prodrugs can be used in a variety of formulations, such as enteric-coated tablets, where the anilinium group can be converted to the corresponding aniline group in the intestine (pH = 6.8) and undergo a proton transfer process to furnish the parent drug and the nontoxic linker [185].

Acyclovir (Figure 9, **13**) is a synthetic purine nucleoside antiviral drug used to treat infections caused by the herpes simplex virus (HSV), herpes zoster (shingles), and varicella-zoster (chickenpox) by inhibiting DNA synthesis and viral replication after conversion to acyclovir triphosphate by cellular or viral enzymes. Acyclovir can be taken orally or by IV administration. The oral efficacy of acyclovir is limited due to its low bioavailability. That is why valacyclovir, a prodrug of acyclovir, is produced [186,187]. Valacyclovir is better absorbed through the gastrointestinal wall with greater absolute bioavailability of 54% compared with acyclovir with just 12–20% due to its poor water solubility and, hence, inefficient permeability through membranes [188,189]. Karaman’s study aimed to design a carrier-linked acyclovir prodrug that is less hydrophilic than the parent drug with higher bioavailability than the valacyclovir prodrug. The acid-catalyzed hydrolysis in some of Kirby’s acid amides might have the potential to be good carriers to acyclovir (Figure 9). DFT calculations at the B3LYP ⁄ 6-31G (d,p) level were made in the gas phase and in ether and water to determine the rate-limiting step. Computation was directed for the elucidation of the transition-state and ground-state structures (reactants, intermediates, and products) for the acid-catalyzed hydrolysis of the prodrugs. Results showed that the rate-limiting step is dependent on the reaction medium, which could be the collapse or the formation of the tetrahedral intermediate. Furthermore, DFT calculations for this approach demonstrated that the rate of hydrolysis is dependent on the substitution on the C–C double bond and on the amide nitrogen substituent, in addition to the structural features of the linker in Kirby’s acid amide moiety. The desirable prodrugs of acyclovir were synthesized using the linker of Kirby’s maleamic acid amide model. In vitro kinetic studies of the designed prodrugs are proceeding to evaluate the ability of the prodrugs to undergo chemical conversion in the physiological environment (37 °C, pH = 2.0 and 6.0 in aqueous medium) and to test the release order of the parent drug, acyclovir [190].

## 7. Conclusions

Enzymes are complex systems, and comprehending their mode of action and the factors responsible for accelerating the speed of the reactions they catalyze is essential in studying their biochemical processes. Molecular modeling and simulation methods are increasingly used to illustrate the mechanisms of enzyme-catalyzed reactions and the determinant factors of specificity and efficiency, which can greatly contribute to the development of drugs and catalysts alike. Despite the advances in experimental studies, a comprehensive understanding of enzyme catalysis can be reached only through the integration of experiments and computer modeling approaches. Several chemical models have been presented by scientists to mimic the high acceleration rates achieved by enzymes, such as Koshland in his “orbital steering” theory, Bruice in his intramolecular cyclization of dicarboxylic semi-esters (NAC), Milstien and Cohen in their gem-tri-methyl lock systems (stereopopulation control), Kirby’s enzyme models of proton transfer in systems containing two heteroatoms, and Menger in his spatiotemporal hypothesis. In addition to the goals achieved by these pioneers in understanding enzyme catalysis, the enzyme models invoked by them can be used as linkers to commonly used drugs for making more bioavailable prodrugs. Understanding the chemistry of many organic mechanisms is effective in the development and design of an efficient chemical device to be used as a prodrug linker where the resulting prodrug can undergo conversion by chemical and not enzymatic means to liberate the active drug in a controlled manner. Based on the above-mentioned enzyme models, different linkers were investigated for the design of a large number of prodrugs to improve the pharmacokinetics and patient compliance of some important currently marketed drugs. These include anti-Parkinson (dopamine), antiviral (acyclovir), antimalarial (atovaquone), anticancer (azanucleosides), antifibrinolytic (tranexamic acid), antihyperlipidemia (statins), vasoconstrictors (phenylephrine), and antihypertension (atenolol). Besides, this approach was used to mask the bitter taste of several drugs, such as antibacterial agents (amoxicillin, cephalexin, and cefuroxime axetil), paracetamol, and guaifenesin. Molecular revolution has changed the vision to the prodrug approach, in particular, from merely a chemical modification to solve problems associated with parent compounds to a modern, promising, safe, and efficacious approach that considers molecular/cellular factors to deliver active small-molecule and biotherapeutics.

## Figures and Tables

**Figure 1 molecules-26-03248-f001:**
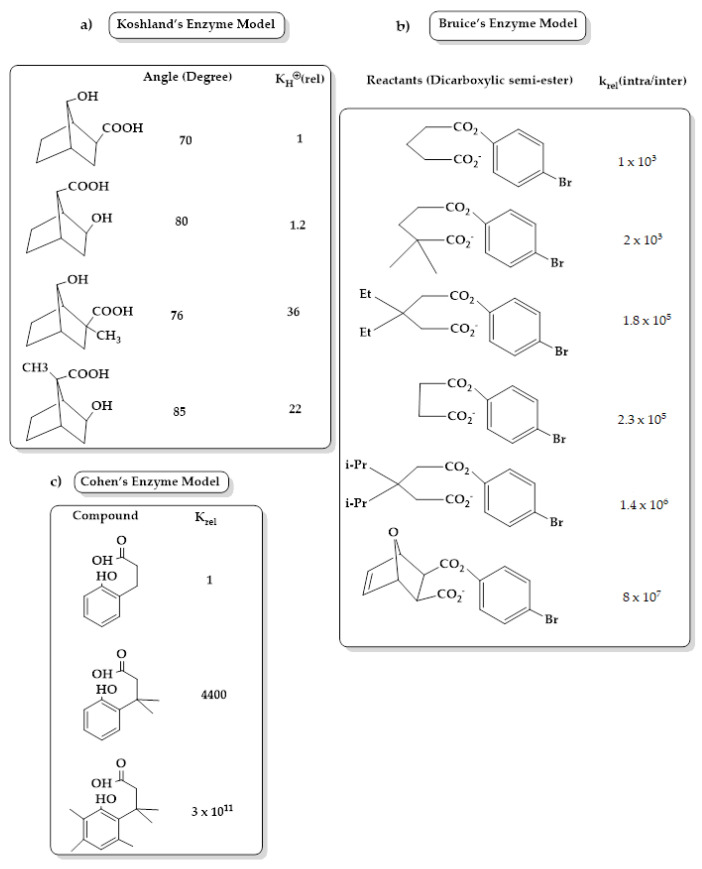
(**a**) Principles of “orbital steering” theory by Koshland, which describe the relative reactivity of lactonization of some hydroxy acids [47]; (**b**) acceleration rates of cyclization of di-carboxylic semi-esters of Bruice’s system [48]; (**c**) Cohen enzyme model of stereopopulation control, which describes the relative reactivity of lactonization of some hydroxyhydrocinnamic acids [47].

**Figure 2 molecules-26-03248-f002:**
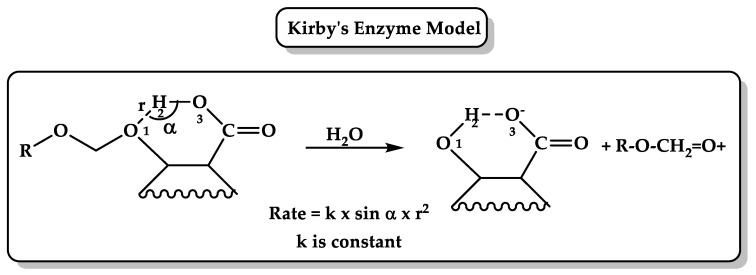
Proton transfer in Kirby’s enzyme model where α is the angle of attack O1H2O3 and r is the distance between the two reacting centers, H2–O1. R is an alkyl or aryl group [57].

**Figure 3 molecules-26-03248-f003:**
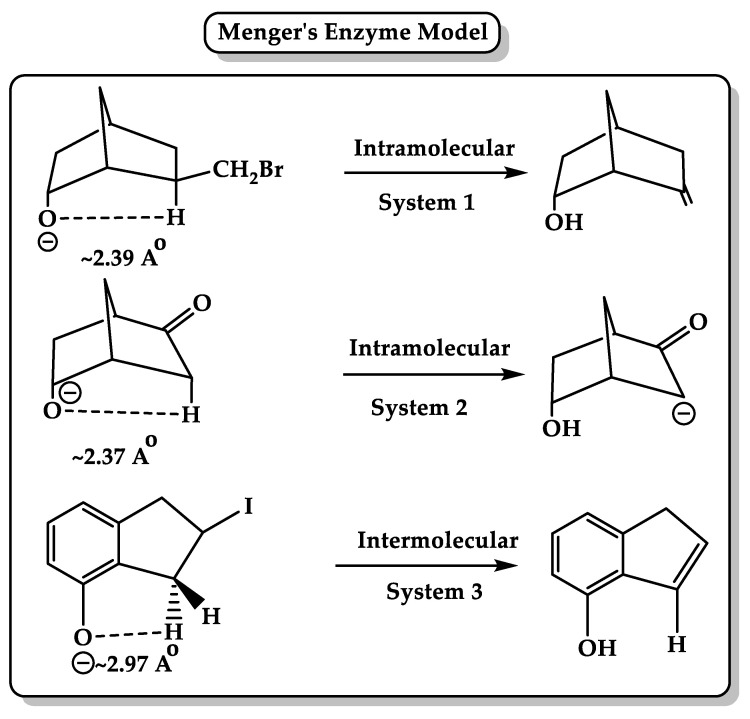
Menger’s enzyme model based on “spatiotemporal hypothesis,” which suggests that intermolecular or intramolecular reaction is determined by the distance between the two reacting centers of the reactant [66].

**Figure 4 molecules-26-03248-f004:**
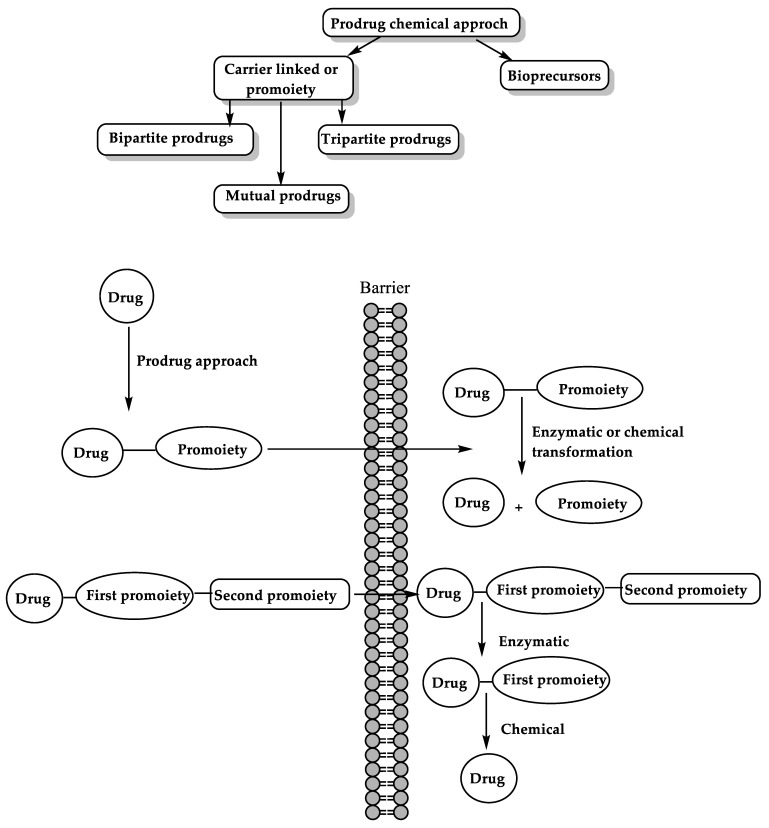
Classification of the chemical prodrug approach and its concept of transformation inside the body.

**Figure 5 molecules-26-03248-f005:**
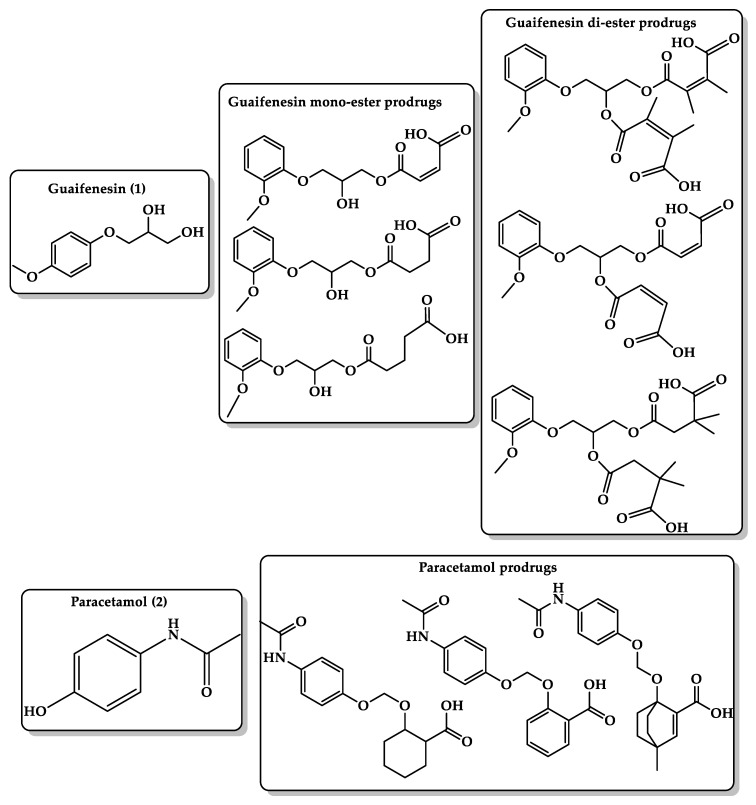
Chemical structures of guaifenesin **(1)** and guaifenesin prodrugs and paracetamol **(2)** and paracetamol prodrugs based on Kirby’s acetal enzyme model.

**Figure 6 molecules-26-03248-f006:**
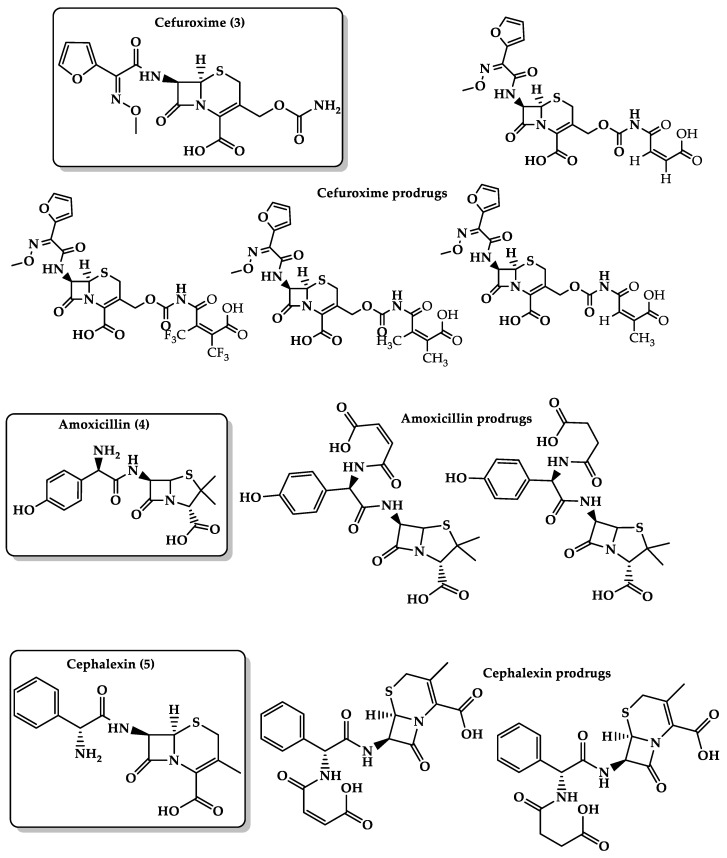
Chemical structures of cefuroxime **(3)** and cefuroxime prodrugs, amoxicillin **(4)** and amoxicillin prodrugs, cephalexin **(5)** and cephalexin prodrugs.

**Figure 7 molecules-26-03248-f007:**
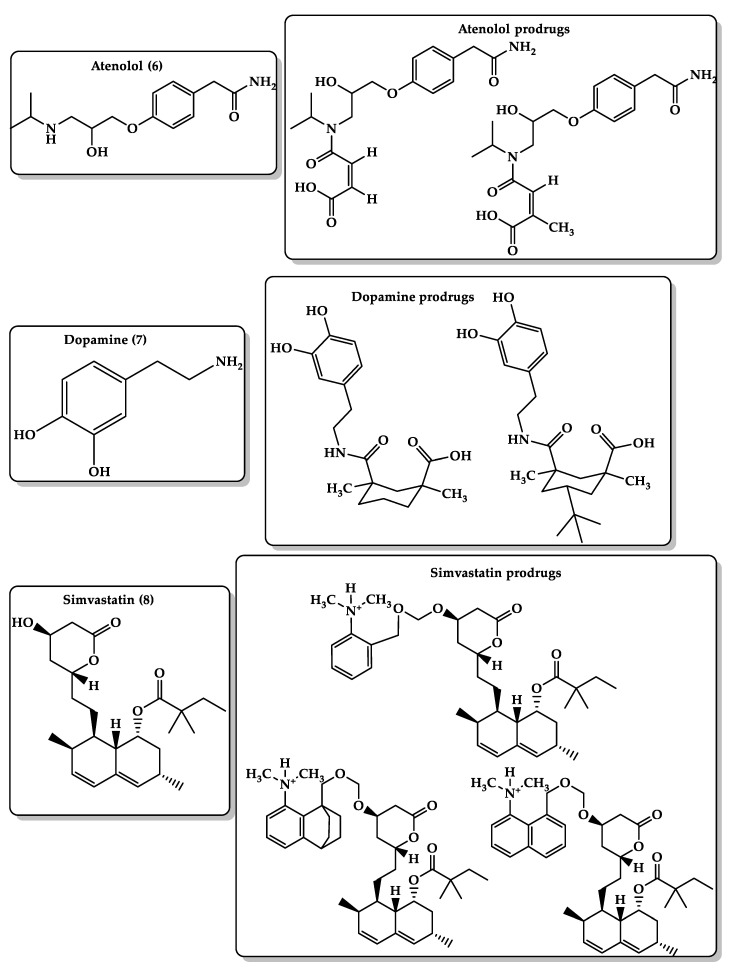
Chemical structures of atenolol **(6)** and its prodrug, dopamine **(7)** and its prodrugs, and simvastatin **(8)** and its prodrugs.

**Figure 8 molecules-26-03248-f008:**
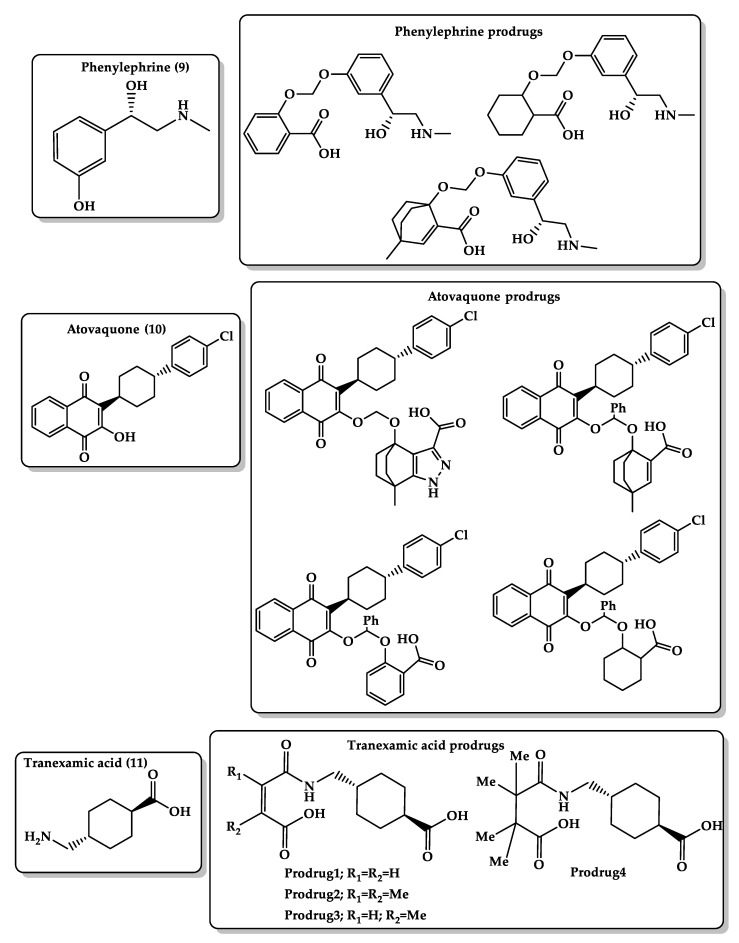
Chemical structures of phenylephrine **(9)** and its prodrugs, atovaquone **(10)** and its prodrugs, and tranexamic acid **(11)** and its prodrugs.

**Figure 9 molecules-26-03248-f009:**
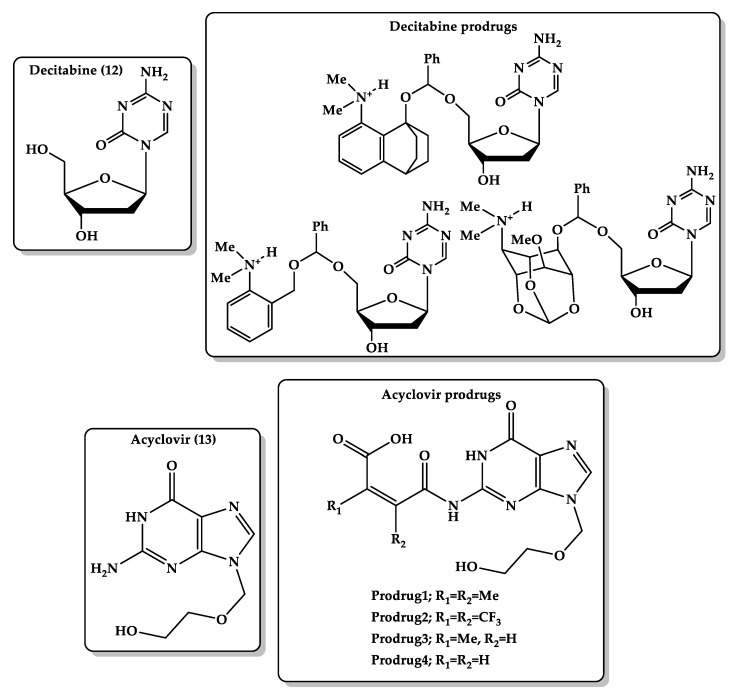
Chemical structures of decitabine **(12)** and its prodrugs and acyclovir **(13)** and its prodrugs.

**Table 1 molecules-26-03248-t001:** Comparison between different computational methods and their applications.

Quantum Mechanics	Molecular Mechanics	Molecular Docking
Used to calculate electronic behavior of atoms and molecules such as electron density [34].	Used for studying physical properties such as structure, energy, and dipole moment by using molecular force fields (FFs) to provide an efficient description of chemical system but cannot calculate electronic behavior [35].	Used to predict interaction between proteins/proteins or proteins/small molecules to evaluate the binding between them. It is used widely in the field of drug screening and design [36].
Used for small molecule, around hundreds of atoms [18].	Used for large molecule, more than ten thousands of atoms [37].	Allows virtual screening of thousands of small molecules [38].
Time and money consuming and requires high computational effort [35].	Time and money consuming with low computational effort [35].	Fast and inexpensive method [39].
	FF approaches cannot describe the formation and cleavage of covalent chemical bonds, while reactive force fields can make such processes accessible [35].	Uses force-field-based fixed dielectric charges for both protein and ligand atoms, which gives false-positive or false-negative protein–ligand binding energy [39]. Ignores many inherent factors underlying ligand–receptor interactions such as solvation [38]. Unfitting target binding site and provides only an approximate assessment of binding affinities [38]. Low accuracy, 65%–75% [38,39]

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
