# Peer review of "Enzyme Models—From Catalysis to Prodrugs"

_molecules, 2021, doi:10.3390/molecules26113248_

Round 1
Reviewer 1 Report
Manuscript ID: molecules_1188367
Dear Authors,
The review manuscript (MS_ID: molecules_1188367) entitled “Enzyme Models-From Catalysis to Prodrugs” by Zeinab Breijyeh and Rafik Karaman summarized the work done on enzyme models and the computational methods, such as Molecular Mechanics (MM) and Quantum Mechanical (QM) methods, used to understand enzyme catalysis and to help in the development of efficient prodrugs. The authors introduce some practical prodrugs designed using enzyme models and the computational methods. There are some lacks of explain of the analytic approaches. Some comments are described below.
Comments:
1) Line 32-34: Few years ago, the enzyme classification has been renewed to 7 groups. Thus, this sentence should be changed to “Enzymes are classified according to the Enzyme Commission (EC) number into seven main groups: oxidoreductases; transferases; hydrolases; lyases; isomerases; ligases and translocases.”
2) Line 56: “… in a synthetic catalyst system.”
3)Line 59-60: Molecular mechanics (MM) and Quantum mechanical (QM) methods have been abbreviated to MM and QM, respectively. Thus, these abbreviations should be used through the manuscript.
4) Most sections are described one or two paragraphs with lengthy sentences. Thus, these sections should be reconstructed to read more easily for readers.
5) In the section 1 to 5, there were no figure including illustrations and scheme. I think that the illustrations can help readers to understand their enzyme models and methods.
6) Line 174: “… is estimated roughly to be 2.8 ” should be corrected to “…. Is estimated roughly to be 2.8 Å”
Author Response
Reviewer 1
- Comment: “Line 32-34: Few years ago, the enzyme classification has been renewed to 7 groups. Thus, this sentence should be changed to “Enzymes are classified according to the Enzyme Commission (EC) number into seven main groups: oxidoreductases; transferases; hydrolases; lyases; isomerases; ligases and translocases.”
Response: Sentence was corrected and new reference was added line 34
- Comment: Line 56: “… in a synthetic catalyst system.”
Response: Corrected line 57.
- Comment: “Line 59-60: Molecular mechanics (MM) and Quantum mechanical (QM) methods have been abbreviated to MM and QM, respectively. Thus, these abbreviations should be used through the manuscript.”
Response: MM and QM abbreviation was used throughout the text
- Comment: Most sections are described one or two paragraphs with lengthy sentences. Thus, these sections should be reconstructed to read more easily for readers.
Response: The paragraphs were reconstructed.
- Comment: “In the section 1 to 5, there were no figure including illustrations and scheme. I think that the illustrations can help readers to understand their enzyme models and methods.”
Response: Some figures have been added through the text illustrating enzyme models.
- Comment: Line 174: “… is estimated roughly to be 2.8 ” should be corrected to “…. Is estimated roughly to be 2.8 Å”
Response: no modification because the letter is in the middle of the text “which is estimated”.
Reviewer 2 Report
This MS is a review article, but the content of the M.S. isn't not sufficient as a review article. Since the major contents are QM/MM in the specific biological catalysts, the author only discussed these small compounds case (such as anti-Parkinson (dopamine), anti-viral (acyclovir), anti-malarial (atovaquone), anticancer (azanucleosides), anti-fibrinolytic (tranexamic acid), anti-hyperlipidemia (statins), vasoconstrictors (phenylephrine), antihypertension (atenolol), antibacterial agents (amoxicillin, cephalexin and cefuroxime axetil), paracetamol, guaifenesin, and others.). I suggest that the authors should widely summarize the current Status and future challenges of QMMM method in the enzyme models.
Author Response
Reviewer 2
Few examples were illustrated in section 6 to satisfy the request by reviewer 2.
Reviewer 3 Report
The revue entitled "Enzyme Models-From Catalysis to Prodrugs" proposes to review computational methods and the theories associated with these methods to understand enzymatic modeling. This approach is indeed essential for designing new drugs from the design of prodrugs. And then to study its behavior in a physiological environment like that of the cell, in particular how it is released and at what speed.
This review shows clearly that there are numerous computational approaches and prodrugs which have been designed and developed aiming to improve the effectiveness of certain drugs and to overcome pharmaceutical/pharmacokinetic obstructions that occur upon drug administration, such as low stability, poor absorption, low solubility, reduced patient compliance.
However, I have some remarks.
Reading the title "enzyme models from catalysis to prodrugs", I was waiting for studies on enzyme applied for prodrug activation and the approaches developed for the evaluation of E-S interactions to release the drugs with enzymes such as hydrolytic enzymes or oxido-reductases. And in these cases, what are the limitations of the approaches ?
In part 2. "computational methods", the authors show that there are many computational methods for enzyme/prodrugs simulations. It would be interesting to show a comparison between them, their applications and limitations... comparison with molecular docking ?
Author Response
Reviewer 3
- Comment: Reading the title "enzyme models from catalysis to prodrugs", I was waiting for studies on enzyme applied for prodrug activation and the approaches developed for the evaluation of E-S interactions to release the drugs with enzymes such as hydrolytic enzymes or oxido-reductases. And in these cases, what are the limitations of the approaches?
Response: A new section was added to discuss enzyme mediated drug activation line 352.
- Comment: In part 2. "computational methods", the authors show that there are many computational methods for enzyme/prodrugs simulations. It would be interesting to show a comparison between them, their applications and limitations... comparison with molecular docking?
Response: A table was added for the comparison between computational methods line 124, in addition to other comparisons through the text from line 100.
Reviewer 4 Report
The work provides a timely review on the topic addressed.
The authors should provide more clearly the link between the computation methods used and the design of pro drugs in several of the case studies presented
Author Response
Reviewer 4
- Comment: The authors should provide more clearly the link between the computation methods used and the design of pro drugs in several of the case studies presented
Response: some paragraphs were modified to clarify the link between computation and prodrug design in the case studies.
Round 2
Reviewer 2 Report
The authors have improve the M.S. This M.S. may be acceptable.
Reviewer 3 Report
The authors answered the questions correctly, so I am in favor of publishing the article in Molecules. The authors answered the questions correctly, so I am in favor of publishing the article in Molecules.